# AUTOREGRESSION WITH SELF-TOKEN PREDICTION

## ABSTRACT

Next-token prediction has been highly effective in language, but its extension to continuous modalities is challenging: regression over correlated latents tends to collapse into near-identity mappings, while discretization via vector-quantized encoders introduces quantization artifacts. Mask-based prediction with diffusion heads mitigates these issues, yet suffers from a train–inference mismatch, inability to use key–value caching, and poor scalability to long sequences. To overcome these limitations, we propose *self-token prediction*, which conditions each token on ground-truth references during training, ensuring consistency with causal inference while avoiding identity collapse. This design supports key–value caching and parallel generation, enabling scalable, high-fidelity synthesis across text, audio, image, and video. Built on this paradigm, OMNIAR unifies heterogeneous modalities in a shared omni-token space, achieving efficient and high-quality generation, including real-time and theoretically endless video generation[1].

## 1 INTRODUCTION

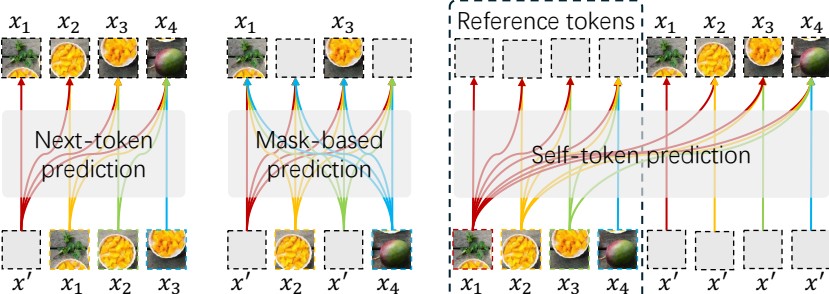

Figure 1: Illustration of next-token prediction, mask-based prediction, and self-token prediction. Self-token prediction leverages reference tokens to enable causal attention while maintaining efficient training for continuous tokens. Here, $x'$ denotes the modality-specific special embeddings, and the loss is only applied to the predicted tokens.

The remarkable progress of large language models (LLMs) has established next-token prediction (NTP) as a powerful and scalable learning paradigm (Kaplan et al., 2020; Xiong et al., 2024). Autoregressively modeling the conditional distribution of the next discrete token has proven both efficient to train and effective at producing fluent, coherent text (Gong et al., 2025). It is therefore natural to ask whether the same paradigm can be extended beyond language to an *omnimodal* setting that jointly supports understanding and generation across text, audio, image and video (Chen et al., 2024c; Downes et al., 2024). In practice, however, a straightforward application of NTP to continuous modalities reveals fundamental mismatches that limit its effectiveness (Lin et al., 2025).

When NTP is applied to discrete token spaces (*e.g.*, text or VQ-VAE (Razavi et al., 2019) visual codes), the training objective is a probabilistic cross-entropy that encourages learning a distribution over a finite vocabulary. Small changes in content can map to different discrete ids, so the model is incentivized to capture high-level, semantic structure rather than simply reproducing low-level inputs. By contrast, continuous latent tokens (*e.g.*, KL-VAE (Ho et al., 2020) features derived from

[1]Project page: `https://anonymous.4open.science/r/OmniAR-168A`.

adjacent frames or neighboring patches) are typically highly correlated: adjacent frames or nearby patches produce very similar latent vectors. Under an NTP-style training objective on continuous values, this similarity turns the learning problem into a numerical regression over nearly identical input–target pairs. In this regime the easiest way to minimize the regression loss is to approximate an identity mapping (*i.e.*, to reproduce the input precisely), which prevents the model from learning useful, higher-level abstractions or reasoning (Huang et al., 2025). In short, the combination of (i) highly correlated continuous latents and (ii) a regression objective creates a strong bias toward identity solutions that is absent when modeling discrete tokens with cross-entropy.

A practical workaround has been to avoid direct autoregressive regression by either discretizing continuous modalities via VQ-VAE (Yu et al., 2023b; Han et al., 2024) or by adopting mask-based prediction (He et al., 2022) with distributional heads. Discretization recovers a cross-entropy objective but at the cost of quantization artifacts and limited representational capacity as modalities scale (for example, to high-resolution images or long videos). More recently, mask-based prediction combined with token-level diffusion heads (Li et al., 2024) has shown substantial promise: by converting value regression into a conditional distribution modeling problem in latent-token space, diffusion heads greatly alleviate the identity-mapping pathology and enable effective training on continuous modalities. Recent works, including D-JEPA·T2I (Chen et al., 2024a;b) and related studies (Chen et al., 2024b; Fan et al., 2025), show that this approach produces high-quality image synthesis and high-resolution text-to-image outputs.

Nonetheless, mask-based approaches introduce their own, practically important limitations. First, they rely on full (dense) attention during training and generation, which prevents using standard key–value caching techniques (Vaswani et al., 2017; Dao et al., 2022; Hooper et al., 2024) and so leads to prohibitive compute and memory costs for long sequences. Second, although the fraction of masked tokens is sampled stochastically during training(see Fig. 1, middle), each forward pass uses only a single sampled mask ratio; consequently, training sees only isolated masking configurations rather than the progressive unmasking dynamics that occur during generation. This sampling granularity produces a form of training–inference mismatch and reduces training efficiency. Moreover, NTP itself (despite being conceptually different) also suffers from sampling inefficiency for long sequences because it generates tokens strictly one-by-one. Together, these factors make both mask-based and naive autoregressive schemes difficult to scale to high-resolution or long-duration content such as long videos.

To address these interlocking problems, we propose the *self-token prediction* paradigm. During training, each token is provided with an explicit *reference token* (the ground-truth continuous latent) and the model learns to predict tokens in a causal decoding order while attending to their corresponding references (see Fig. 1, right). Two key properties arise from this design. First, by conditioning on ground-truth references during training the formulation avoids the identity-mapping collapse that plagues plain regression: the model learns to map from context to a distribution anchored by the reference rather than simply copying the input. Second, because training explicitly mirrors the causal generation process, self-token prediction ensures consistency between training and inference. This consistency enables safe use of key–value caching and supports decoding strategies that generate many tokens in parallel (for example, frame-by-frame or neighborhood-based generation), thereby addressing the sampling inefficiency of both mask-based methods and one-by-one autoregression without degrading quality. In other words, every training pass under self-token prediction reproduces a complete generation step (via references), rather than exposing the model to only a single, randomly sampled masking fraction as in prior mask-based training (Chen et al., 2024a; Li et al., 2023b; 2024).

Built on these ideas, we present OMNIAR, an omnimodal foundation model that aligns heterogeneous modalities into a shared omni-token space. OMNIAR preserves the strengths of next-token prediction for discrete text understanding while using self-token prediction to enable high-fidelity, scalable generation of continuous modalities (image, audio, video). The combination of token-level distributional heads (*e.g.*, diffusion (Ho et al., 2020) or flow matching (Lipman et al., 2022)) and self-token prediction both mitigates identity collapse and unlocks efficient long-sequence synthesis: OMNIAR can leverage key–value caching for causal decoding and supports batch-generation strategies (*e.g.*, per-frame or local-neighborhood decoding) that make long and even real-time video generation tractable. Extensive experiments show that OMNIAR attains strong fidelity across modalities and scales gracefully to long sequences, establishing self-token prediction as a practical and versatile foundation for next-generation omnimodal models.

## 2 BACKGROUND

Autoregressive language models have achieved remarkable scalability by unifying diverse NLP tasks under next-token prediction (Brown et al., 2020; OpenAI, 2023; Touvron et al., 2023; Bai et al., 2023), inspiring extensions to images and videos via VQ-VAE discretization (Ramesh et al., 2021; Esser et al., 2021) and, more recently, continuous tokenizers (Chen et al., 2024a; Li et al., 2024; Fan et al., 2024), refined objectives (Chang et al., 2022; Tian et al., 2024; Pang et al., 2024), and expanded vocabularies (Tang et al., 2024; Agarwal et al., 2025). While these approaches improve fidelity and scalability, they remain limited in bridging discrete text with continuous visual and auditory data. Multimodal LLMs address cross-modal understanding by integrating visual encoders (Li et al., 2023a; Zhu et al., 2023; Dai et al., 2023; Liu et al., 2024b) or leveraging pretrained vision–language features (Radford et al., 2021; Zhai et al., 2023), yet they remain text-centric, often separating understanding from generation and relying on diffusion or complex architectures (Dong et al., 2023; Ge et al., 2023; Team, 2024; Chen et al., 2025; Wang et al., 2024b). Omnimodal models such as OmniVL (Wang et al., 2022), Emu (Sun et al., 2023b), Show-O (Xie et al., 2024), Ming-Omni (AI et al., 2025), and Omni-R1 (Zhong et al., 2025) expand coverage to audio and video and support reasoning or multi-branch routing, but are still constrained by discretization bottlenecks, pretrained encoders, or inefficient sampling. In contrast, OMNIAR introduces a unified omnimodal architecture based on self-token prediction, bridging discrete and continuous modalities within a single Transformer head without external encoders, enabling scalable, high-fidelity generation across text, image, audio, and video.

## 3 AUTOREGRESSION WITH SELF-TOKEN PREDICTION

As illustrated in Fig. 3, building upon denoising joint embedding predictive architectures (Chen et al., 2024a), OMNIAR introduces *self-token prediction* as a replacement for conventional mask-based objectives. This novel design mitigates the train–inference discrepancy and enables efficient key–value (KV) caching, thereby addressing key limitations of prior approaches (Chen et al., 2024b; Fan et al., 2025).

### 3.1 ARCHITECTURE

**Backbone.** OMNIAR adapts QWEN3 (Xu et al., 2025) as its backbone, a decoder-only Transformer architecture known for its performance and public availability. The backbone employs RMSNorm (Zhang & Sennrich, 2019) for normalization, SwiGLU (Shazeer, 2020) for activation, RoPE (Su et al., 2024) for positional encoding, and GQA (Ainslie et al., 2023) for efficient KV caching. Following recent advances in large-scale generative modeling (Esser et al., 2024), QWEN3 also integrates QK-Norm (Dehghani et al., 2023) within each attention block, substantially stabilizing optimization.

**Mixture-of-Transformers.** To jointly support understanding and generation, we extend the backbone into a Mixture-of-Transformers (MoT) architecture (Deng et al., 2025; Liao et al., 2025). Two parallel Transformer stacks are instantiated: one specialized for understanding and the other for generation, both initialized from pretrained QWEN3 weights. Information exchange between the stacks occurs through joint attention, enabling cross-modal reasoning without degrading task specialization. This design improves convergence stability while ensuring balanced capacity allocation across modalities.

**Multimodal Input Encoding.** For continuous modalities (images, audio, and video), we employ a lightweight encoder based on a compact Vision Transformer (Yu et al., 2023a) and adapt the grouped causal attention used in the backbone. Unlike CLIP (Radford et al., 2021) or SigLIP (Zhai et al., 2023), which rely on large-scale pretraining, our encoders are trained from scratch to project modality-specific latents into the shared *omni-token* space via the JEPA loss (Assran et al., 2023) (*i.e.*, Eq. 4). Text inputs are directly embedded through an embedding layer. All tokens are hard-routed within the backbone, enabling flexible cross-modal interactions while preserving efficiency.

**Multimodal Output Decoders.** To enable generation across modalities, we introduce lightweight ($\sim$300M parameters) modality-specific denoising heads, trained jointly with the backbone. Each head adopts flow matching (Liu et al., 2022) to render the target modality conditioned on backbone features $z_i$. For audio and image synthesis, we use denoising MLPs (Li et al., 2024), which are more

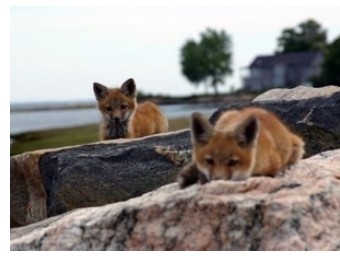

**Two foxes** *resting* on **large rocks**. The foreground features a fox *lying down*, with its head resting on the rock. The second fox is *sitting upright* on a rock in the midground, *looking directly* at the camera. Both foxes have a reddish-brown fur, with darker shades around their eyes and ears.
In the background, there is **a body of water**, possibly a lake or a calm sea. Beyond the water, there is a blurred landscape that includes **a few trees** and **a house or building**. The sky is overcast, with a soft, diffused light that gives the scene a serene and peaceful atmosphere.

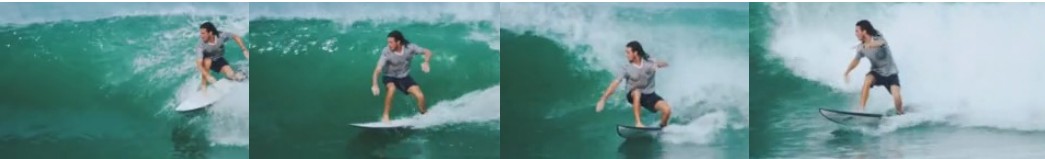

A man is surfing on a white surfboard with a black stripe on the bottom, riding a large, green wave. He is wearing a striped shirt and black shorts. The man maintains his balance by adjusting his stance and using his arms for stability. The wave behind him is large and crashing, creating a dynamic and energetic background. The man continues to surf smoothly, making slight turns and adjustments to his position on the board. The water is a vibrant green, indicating a clear and sunny day. The man skillfully maneuvers through the wave, showcasing his surfing skills. The man maintains his balance and control over the surfboard as he rides the wave.

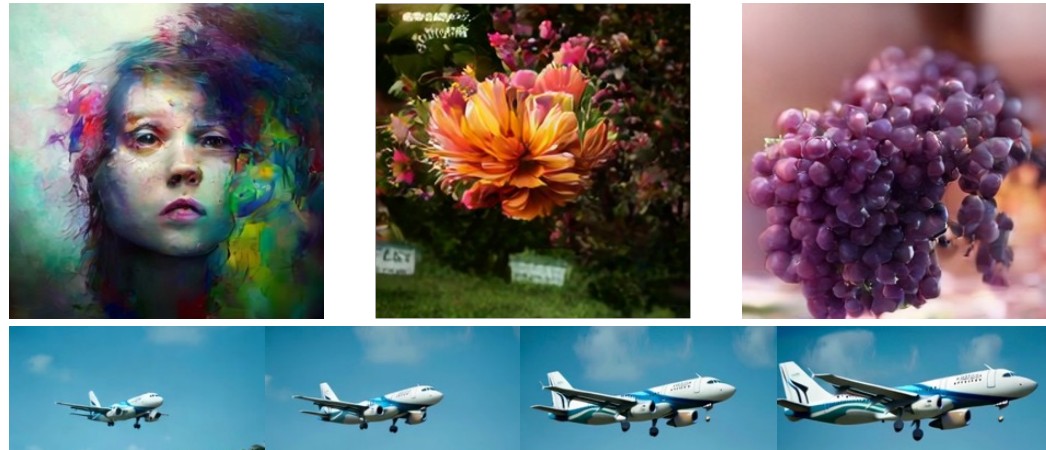

Figure 2: Qualitative results of OMNIAR. Top row: image captioning; second row: video captioning; third row: text-to-image generation; bottom row: video generation.

computationally efficient and operate at a fine-grained token or patch level. For video generation, we adopt a denoising Transformer (Chen et al., 2023b) that reconstructs frames in parallel, capturing temporal coherence.

**Positional Encoding.** While RoPE (Su et al., 2024) is effective for discrete text, it suffers when applied to continuous spatiotemporal modalities. To address this, we employ VoPE (Chen et al., 2024b), a RoPE-inspired scheme tailored for vision and audio, further integrated with MM-RoPE (Yuan et al., 2025) for unified multimodal representations. Each token is associated with an 8-dimensional meta-vector encoding *(global position ID, frame index, height, width)*, with each scalar expanded into two dimensions. Tokens from the same modality share a global position ID, mitigating long-context interference in text generation (Yang et al., 2025). This meta-information is projected into the embedding dimension and incorporated multiplicatively, ensuring modality-consistent grounding across the entire sequence.

### 3.2 SELF-TOKEN PREDICTION

Conventional next-token prediction often degrades continuous generation (*e.g.*, images, video), since previously generated tokens lack precise positional cues for the current token. Prior works attempt to alleviate this by reinjecting target-aware position embedding (Yu et al., 2024), but this

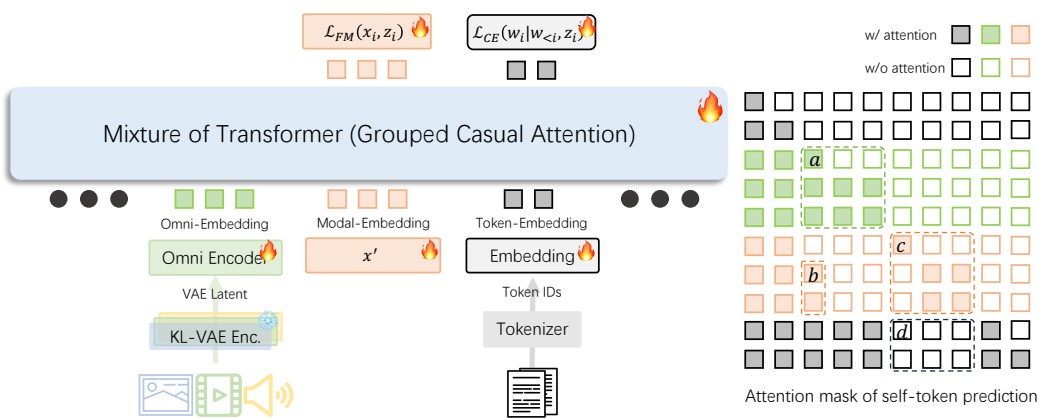

Figure 3: OMNIAR: An omnimodal Transformer that bridges discrete text and continuous modalities through self-token prediction. Special tokens $x'$ are used as modality-specific embeddings for prediction.

remains insufficient. We instead propose *self-token prediction*, a paradigm that avoids dependence on autoregressively emitted outputs, as shown in Fig. 1.

Prediction tokens are initialized with a special embedding $x'$, analogous to masked prediction but fully compatible with autoregressive decoding. The model maintains two parallel streams: *reference tokens*, derived from the ground truth and processed with *grouped causal attention* (field $a$ in Fig. 3); and *prediction tokens*, which attend freely within their group (field $c$) and causally to preceding reference tokens (field $b$). All other tokens attend exclusively to reference tokens, ignoring prediction tokens (field $d$). In other words, prediction tokens are never used as context for modeling or generating the remaining tokens. This asymmetric attention design enforces a faithful generative trajectory while remaining fully compatible with causal attention and key–value caching.

**Grouped causal attention.** Grouped causal attention abstracts token-generation strategies across modalities while aligning training and inference. For images, we adopt a next-neighbor scheme: tokens are ordered by Manhattan distance from a randomized point. We then determine the number of groups $G$ via a logarithmic law:

$$G = \left\lfloor \log_b\big(1 - a\,n\big) \right\rfloor, \tag{1}$$

where $n$ is the total number of image tokens, and $a, b \in \mathbb{R}^+$ are tunable coefficients (with clamping to ensure validity). In practice, we set $a = 0.08$ and $b = 1.10$, which yields a reasonable growth of group count as the number of tokens increases. As shown in Fig. 4, compared to strictly token-wise grouping, row-/column-based grouping or next-neighbor grouping (He et al., 2025), Eq. 1 achieves a better trade-off: it maintains high generation quality while requiring fewer groups. Fewer groups directly reduce the number of forward passes at inference time.

Unlike uniform partitioning of $n$ tokens into $G$ groups, we argue that early-stage groups should be smaller, since little context is available, while later groups can be larger due to richer context. Thus, group sizes follow a cosine allocation:

$$\gamma_\tau = \tfrac{1}{2}\Big(1 + \cos\Big(\pi\,\tfrac{\tau}{G}\Big)\Big), \qquad n_\tau = \lfloor \gamma_\tau\,n \rfloor, \quad \tau = 1, \ldots, G, \tag{2}$$

with the final group set to absorb the remainder such that $\sum_{\tau=1}^{G} n_\tau = n$. Fig. 4 visualizes the per-step token allocation when generating a $256 \times 256$ image, showing that early steps generate fewer tokens and later steps more.

The distance-ordered list is thus partitioned into $G$ contiguous groups $\{n_\tau\}$, with full intra-group attention and causal inter-group attention. For video, all tokens within a frame form one group (next-frame generation). For audio, strict temporal order is preserved (next-sample generation).

### 3.3 TRAINING OBJECTIVE

**Cross-Entropy Loss.** For discrete text generation, we adopt the standard negative log-likelihood:

$$\mathcal{L}_{\text{CE}} = -\sum_{i=1}^{N} \log p(w_i \mid w_{<i}, z_i), \tag{3}$$

where $w_i$ is the $i$-th token, $w_{<i}$ the prefix, and $z_i$ the associated multimodal feature. This ensures accurate autoregressive grounding in language.

**JEPA Loss.** To encourage modality alignment, we extend JEPA (Assran et al., 2023) to token-level embeddings:

$$\mathcal{L}_{\text{JEPA}} = \sum_{i=1}^{n_\tau^{\text{pred}}} \left\| F(z_i) - y_i \right\|_1, \tag{4}$$

where $z_i = E_{\text{ctx}}(x_i)$ and $y_i = E_{\text{trg}}(x_i)$. The context encoder $E_{\text{ctx}}$ observes partial tokens via grouped causal attention, while the target encoder $E_{\text{trg}}$ observes full inputs via full (dense) attention. Intuitively, Eq. 4 enforces the backbone $F$ to produce features that are as close as possible to those obtained with full attention, thereby enhancing the model's ability to learn and understand representations. During training, $E_{\text{ctx}}$ is optimized jointly with the backbone $F$, while $E_{\text{trg}}$ is updated via EMA of $E_{\text{ctx}}$ (Assran et al., 2025; Bardes et al., 2023; Chen et al., 2024a).

**Flow Matching Loss.** For continuous token prediction, we leverage flow matching (Lipman et al., 2022). Given noise $\epsilon \sim \mathcal{N}(0, I)$ and interpolation $t \sim \mathcal{U}(0, 1)$, we perturb inputs as

$$x_i^t = t x_i + (1 - t)\epsilon.$$

The model learns a velocity field $v_\theta$ via:

$$\mathcal{L}_{\text{FM}} = \mathbb{E}_{t,\epsilon}\left[\left\| v_\theta(x_i^t, t, z_i) - (x_i - \epsilon) \right\|_2^2\right], \tag{5}$$

with $v_\theta$ parameterized by a denoising MLP (Li et al., 2024) or a denoising Transformer (Chen et al., 2023b).

**Final Objective.** The overall loss integrates all components:

$$\mathcal{L} = \mathcal{L}_{\text{FM}} + \lambda_{\text{CE}}\, \mathcal{L}_{\text{CE}} + \lambda_{\text{JEPA}}\, \mathcal{L}_{\text{JEPA}}, \tag{6}$$

where $\lambda_{\text{CE}}$ balances linguistic grounding and generation, while $\lambda_{\text{JEPA}}$ regularizes multimodal alignment. Following Fan et al. (2025), all losses are applied to prediction tokens, improving efficiency by avoiding unnecessary reconstruction. We further observe that setting $\lambda_{\text{CE}}$ and $\lambda_{\text{JEPA}}$ too high significantly degrades the quality of image and video generation. Therefore, their values are typically kept around the order of $0.001$, which maintains strong generative performance without noticeably impairing understanding ability.

### 3.4 EFFICIENT OMNIMODAL TRAINING

A key challenge in multimodal training is maintaining stability across heterogeneous tasks. A naive approach requires all nodes to sample the same task type in each iteration so that computation graphs remain identical across workers. Although this improves computational efficiency, we observe severe instability when task distributions are highly imbalanced: the model frequently switches between tasks, loss curves oscillate sharply, and in extreme cases training may even diverge.

To mitigate this, we adopt a more robust strategy: each node samples tasks independently at every iteration, while performing an additional dummy forward pass on unmatched tasks to maintain graph consistency. Although this mixed-task scheme stabilizes training, these dummy passes introduce zero gradients, causing the gradients from rare tasks to be diluted during all-reduce averaging.

To address this gradient underrepresentation, we introduce a dynamic scaling mechanism inspired by federated learning (Chen et al., 2023a). Suppose the $i$-th node processes $s_i$ samples, and let

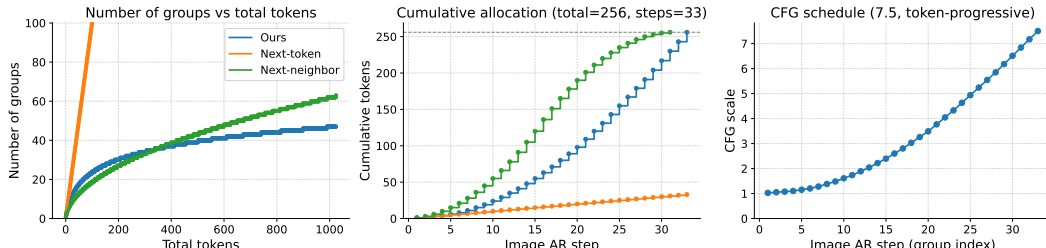

Figure 4: Groups allocated via Eq. 1. Low-resolution images use smaller groups, similar to next-token prediction, while high-resolution images have slower group growth than next-neighbor and next-token, enabling efficient generation.

$S = \sum_{i=1}^{m} s_i$ be the total samples across the $m$ non-empty nodes. Denote by $W$ the total number of nodes (world size). The scaling factor applied to the loss of each modality on node $i$ is defined as:

$$\text{scale}(s_i) = \frac{s_i}{S} \cdot \frac{W}{m}. \tag{7}$$

This formula ensures that each modality's loss is scaled according to both its local sample count and the global task distribution. In practice, we compute this factor independently for every modality and multiply it with the corresponding loss before backward. Empirically, this scaling strategy significantly improves convergence under imbalanced modality distributions, ensuring that gradients from rare modalities are preserved rather than suppressed during all-reduce averaging.

## 4 SAMPLING WITH SELF-TOKEN PREDICTION

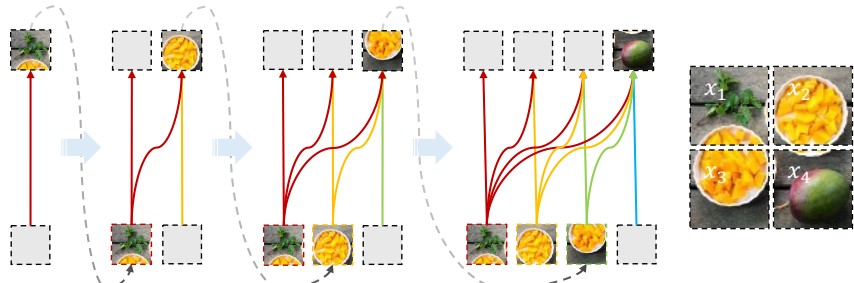

Figure 5: The progressive generation via self-token prediction. Unlike next-token prediction, during inference, self-token prediction uses special embeddings as input and requires updating the key–value cache with the newly generated tokens.

### 4.1 PROGRESSIVE SAMPING WITH CACHE REFRESH

As illustrated in Fig. 5, inference uses strictly causal attention and preserves the grouping strategy employed during training. Relative to conventional next-token decoding, our procedure introduces two key modifications.

**Parallel Token Prediction.** Instead of feeding previously generated tokens directly into the model, we introduce a *special embedding* as input when predicting the current tokens. This design allows the model to generate multiple tokens in parallel within a group, thereby mitigating the strict token-by-token constraint of standard autoregressive decoding.

**Cache Refresh with Omni-Tokens.** To ensure the correctness of KV caching, each newly generated token must replace the corresponding special embedding used during its prediction. Otherwise, the cache would contain representations associated only with special embeddings, which—as discussed in Sec. 3.2—never serve as valid context. To resolve this mismatch, the generated tokens are re-encoded via the omni-encoder to produce omni-tokens, which are then used to update the cache. Importantly,

this cache update is performed jointly with token prediction, avoiding additional forward passes and thereby preserving inference efficiency.

## 4.2 TOKEN-LEVEL SAMPLING

Since we employ flow matching as the rendering head, many techniques originally developed to improve sampling quality in diffusion models can be naturally adapted to our framework. In particular, we highlight two strategies: Classifier-Free Guidance (CFG) (Ho & Salimans, 2022) and time reparameterization (Gao et al., 2024; Esser et al., 2024).

**Classifier-Free Guidance.** During inference, we modulate the guidance strength progressively according to the proportion of tokens already generated. Let $n$ denote the total number of tokens to generate and $N_\tau = \sum_{j=1}^{\tau} n_j$ the cumulative number of tokens after group $\tau$ (with group sizes $n_\tau$ defined in Eq. 1). For a target CFG scale $s > 1$, the per-group weight $w_\tau$ is defined as a linear function of the generation progress:

$$w_\tau = 1 + \frac{N_\tau}{n}(s-1), \qquad \tau = 1, \ldots, G. \tag{8}$$

Early groups, which correspond to a small $N_\tau$, are assigned weaker guidance to encourage sample diversity, while later groups approach the target scale $s$ as more context becomes available, as shown in Fig. 4.

For audio and video synthesis, we instead adopt a constant schedule $w_\tau = c$ with $c > 1$, which we find yields more stable results in temporal domains.

**Time Reparameterization.** We additionally apply a time reparameterization:

$$t' = \frac{t}{t + \Delta t - \Delta t\, t}, \qquad \Delta t > 0, \tag{9}$$

where $t \in [0, 1]$ is the normalized original time variable, and $t'$ is the transformed value. This reparameterization can be viewed as a smooth rescaling that adjusts the effective progression of time as a function of both $t$ and $\Delta t$. Empirically, moderate $\Delta t$ values sharpen and stabilize generations, while excessively large values oversuppress variation and lead to blurry outputs.

In flow-matching generative models (Esser et al., 2024), $\Delta t$ is typically set within $[5, 7]$ to achieve strong perceptual quality. In contrast, our *token-level* formulation is substantially more sensitive: setting $\Delta t$ in the range 1.0–1.5 consistently improves quality, whereas $\Delta t > 2$ noticeably degrades performance.

Ablations on both the CFG scheduling strategy and the time reparameterization factor are provided in the Appendix.

## 5 EXPERIMENTS ANALYSIS

Detailed experimental settings are provided in App. C. Beyond benchmarks, OMNIAR-AVI further demonstrates emerging cross-modal capabilities, including audio-driven image animation, automatic video soundtracking, and image-to-video synthesis. Additional examples are provided on the *project page*, while representative qualitative results are shown in Fig. 2.

### 5.1 EFFICIENCY OF SELF-TOKEN PREDICTION

We investigate whether self-token prediction can serve as a principled alternative to mask-based prediction on ImageNet (Deng et al., 2009), using OMNIAR with ViT-B as both omni-encoder and backbone. Two training paradigms are compared: (i) mask-based prediction with full attention and (ii) self-token prediction with grouped causal attention. To unify generation and classification, we randomly reverse the order of image and label tokens with probability 0.3; the label branch can be regarded as a special prompt encoded via dedicated embeddings. Other training settings follow (Chen et al., 2024a).

| Method | #Params | #Epochs | FID↓ w/o CFG | IS↑ w/o CFG | FID↓ w/ CFG | IS↑ w/ CFG | Acc. |
|--------|---------|---------|--------------|-------------|-------------|------------|------|
| MAR-B (2024) | 208M | 800 | 3.48 | 192.4 | 2.31 | 281.7 | N/A |
| D-JEPA-B (2024a) | 212M | 1400 | 3.40 | 197.1 | 2.08 | 320.9 | N/A |
| MaskGIT (2022) | 227M | 300 | 6.18 | 182.1 | - | - | N/A |
| VAR-d16 (2024) | 310M | - | - | - | 3.30 | 274.4 | N/A |
| MAGE (2023b) | 230M | 1600 | 6.93 | 195.8 | - | - | N/A |
| DiT-XL (2023) | 675M | 1400 | 9.62 | 121.5 | 2.27 | 278.2 | N/A |
| Mask-based pred. | 212M | 400 | 3.83 | 194.2 | 2.43 | 289.3 | 77.8 |
| Self-token pred. | 212M | 400 | 4.01 | 185.2 | 2.78 | 272.2 | 77.5 |
| Mask-based pred. | 212M | 800 | 3.50 | 200.2 | 2.15 | 318.4 | 78.0 |
| Self-token pred. | 212M | 800 | **3.38** | **203.1** | **2.03** | **323.0** | **78.2** |

| Method | Acc. |
|--------|------|
| *Representation models* | |
| I-JEPA (2023) | 72.9 |
| MAE (2022) | 68.0 |
| MAGE (2023b) | 74.7 |
| DINO (2021) | 72.8 |
| *Generative models* | |
| BigBiGAN (2019) | 56.6 |
| MaskGIT (2022) | 57.4 |
| ViT-VQGAN (2021) | 65.1 |
| D-JEPA (2024a) | 46.8 |
| Ours | 69.1 |

(a) System-level comparison on conditional generation and top-1 accuracy of classification.

(b) Top-1 accuracy of linear-prob.

Table 1: Comparison on ImageNet benchmarks. For a fair comparison, we only report results from methods with comparable model sizes.

| Method | #Params | Image | Video | KV Cache | Cnt. Tokens | GenEval↑ | VBench↑ | BLIP↑ |
|--------|---------|-------|-------|----------|-------------|----------|---------|-------|
| *Base scale model* | | | | | | | | |
| Show-O (2024) | 1.3B | ✓ | × | ✓ | × | 0.53 | N/A | - |
| OMNIAR-I | 1.3B | ✓ | × | ✓ | ✓ | 0.56 | N/A | 60.2 |
| *Large Scale model* | | | | | | | | |
| UniFluid (2025) | 2B | ✓ | × | × | ✓ | 0.59 | N/A | - |
| D-JEPA·T2I (2024b) | 2.6B | ✓ | × | × | ✓ | 0.66 | N/A | - |
| OMNIAR-IV | 2.9B | ✓ | ✓ | ✓ | ✓ | 0.62 | 75.12 | 71.3 |
| Lumos-1 (2025) | 3.6B | ✓ | ✓ | ✓ | × | 0.66 | 78.32 | - |
| *Huge scale model* | | | | | | | | |
| OMNIAR-AVI | 5.5B | ✓ | ✓ | ✓ | ✓ | **0.67** | **81.54** | **80.5** |
| EMU3 (2024b) | 8B | ✓ | ✓ | ✓ | × | 0.66 | 80.96 | 79.6 |
| Chameleon (2024) | 7B | ✓ | × | ✓ | × | 0.39 | N/A | 54.1 |
| Transfusion (2024) | 7B | ✓ | × | × | ✓ | 0.63 | N/A | - |

Table 2: Overall performance comparison. OMNIAR supports key–value caching to ensure efficient inference, while preserving continuous tokens to maintain representation quality.

Results in Tab. 1a show that mask-based prediction achieves lower FID at 400 epochs (3.83 vs. 4.01) due to richer context under full attention, but self-token prediction surpasses it after sufficient training (3.38 vs. 3.50). This demonstrates two points: (i) the train–test mismatch inherent in masking ultimately limits performance, and (ii) despite less context, causal attention can be fully exploited with longer training, yielding superior generative quality. In contrast, classification accuracy converges quickly for both paradigms (∼78% at 400 epochs), suggesting that understanding tasks are easier to learn than generation, which should thus receive higher training emphasis. Furthermore, linear probing results in Tab. 1b confirm the high quality of omni-modal representations (69.1% on ImageNet), surpassing generative counterparts and comparable to specialized representation models.

## 5.2 PERFORMANCE OF OMNIAR

We then evaluate multimodal generation. For images, we report GenEval (Ghosh et al., 2023)[2]; captioning is assessed with BLIP (Li et al., 2022) on COCO, and videos are benchmarked by VBench (Huang et al., 2024).

As summarized in Tab. 2, OMNIAR scales effectively: text-to-image GenEval improves from 0.56 to 0.67 and visual understanding from 60.2 to 80.5, both reaching state-of-the-art. In video generation, OMNIAR-IV matches Lumos-1 (Yuan et al., 2025) with fewer parameters (2.9B vs. 3.6B), while the 5.5B OMNIAR-AVI achieves parity with EMU3 (Wang et al., 2024b) (8B).

## 6 CONCLUSION

We introduce OMNIAR with *self-token prediction*, an omni-model capable of comprehending and generating images, videos, and audio. By progressively scaling in both model capacity and modality coverage, OMNIAR establishes a versatile foundation for future omni-model applications, such as synchronized audio–video synthesis, interactive editing, and speech-driven content generation.

---

[2]Scores are obtained on official prompts without rewriting or DPO/reward fine-tuning.

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

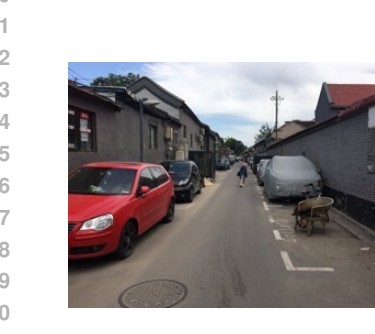

A street scene in what appears to be a residential area. The photograph is taken from a slightly elevated angle, giving a wide-angle view of the street.
On the **left side** of the street, there **is a red car** parked close to the curb. **Behind** the red car, there are **several other vehicles** parked along the street. The buildings on this side of the street are made of brick and have a traditional architectural style. Some of the buildings have signs and windows, indicating that they are likely residential homes.
On the **right side** of the street, there is **a large, gray car** parked along the curb. In front of this car, **a person** is walking down the street, heading towards the camera. The person is wearing casual clothing. There appears to be **a small, wooden chair** placed on the sidewalk. The chair is empty and seems to be a makeshift object.
The street itself is paved and appears to be relatively narrow. There are **white parking lines** on the road, indicating designated parking spaces. The street is lined with buildings on both sides, creating a somewhat enclosed feel.
In the background, there are more buildings, some with **red roofs.** The sky is partly cloudy, suggesting it might be a day with mixed weather conditions.

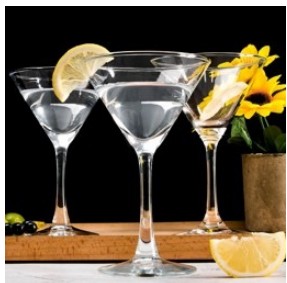

A still life arrangement featuring **three martini glasses** filled with a clear liquid, likely a martini. Each glass is garnished with a lemon slice, adding a pop of color to the composition. The glasses are placed on a wooden surface, which provides a rustic contrast to the sleek, modern design of the glasses.
In the background, there is a vase containing a vibrant yellow sunflower, which adds a touch of natural beauty to the scene. The sunflower is positioned to the right of the glasses, partially out of focus, creating a bokeh effect that draws attention to the glasses and their contents.
To the left of the glasses, a small of a lemon, cut in half, revealing the juicy interior. Beside the lemon, there are a few black olives, adding a classic element to the composition.
The overall style of the image is Photographic, specifically using a standard lens style. The lighting is soft and even, highlighting the clarity and elegance of the martini glasses. The background is a solid black color, which further emphasizes the subjects in the foreground.

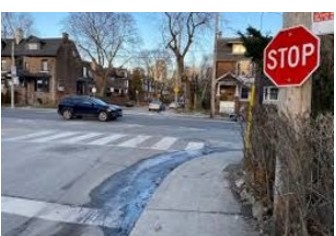

A quiet residential street with **a stop sign** on the right side. a car is driving down the street, which is lined with houses and trees. the sky is clear, suggesting a calm, sunny day.

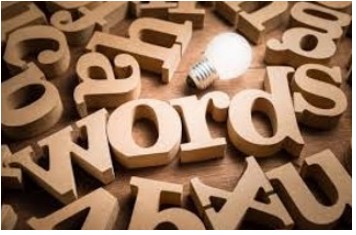

A collection of wooden letters scattered across a wooden surface. The letters are in various orientations, creating a somewhat chaotic arrangement. Among the wooden letters, there is **a single light bulb**, which stands out due to its bright, glowing appearance. The light bulb is positioned centrally in the image, drawing attention to it. The overall style of the image is photographic, with a focus on the wooden letters and the light bulb. The lighting in the image is soft, with the light bulb providing a focal point. The background is a solid wooden surface, which complements the wooden letters.

Figure 6: Caption generated by OMNIAR-I.

# A    USE OF LARGE LANGUAGE MODELS

During the writing of this paper, we utilized large language models (LLMs), such as ChatGPT, to refine the wording of certain sections, primarily within the *Introduction*. No other parts of the manuscript were directly modified using LLMs. All experimental ideas, designs, and analyses were entirely conceived and executed by the authors without any assistance from LLMs. Therefore, this work is entirely original in terms of its scientific contributions and experimental methodology.

# B    MORE EXAMPLES

**Caption.**    In Fig. 6, we present additional examples of captions generated by OMNIAR-I. Remarkably, even with a 0.6B-parameter backbone, OMNIAR is able to accurately capture numerical

relationships among objects, textual content, and spatial configurations within images. The model further demonstrates robustness in describing complex and cluttered scenes, while being capable of producing both detailed and concise captions depending on the context.

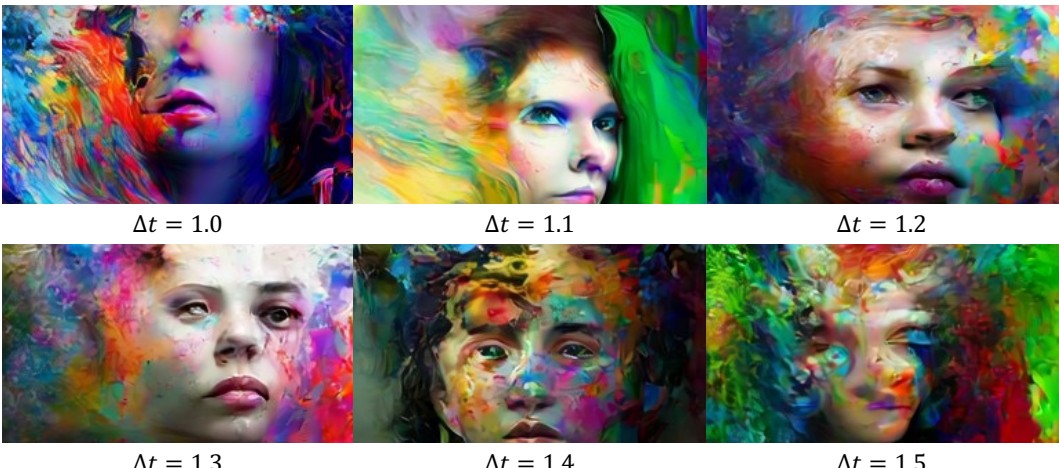

Figure 7: The influence of time shifting factor $\Delta t$.

**Time shifting factor.** Unlike diffusion or flow-matching models, the denoising head in self-token prediction is highly sensitive to the time shifting factor. As shown in Fig. 7, when no adjustment is applied (*i.e.*, $\Delta t = 1.0$), the generated images tend to be overly abstract, lacking fine details and often failing to render complete objects. Introducing a moderate shift, with $\Delta t$ in the range of 1.1–1.2, substantially enhances visual details and improves object integrity. However, increasing $\Delta t$ beyond this range severely compromises the clarity of the generated images, leading to significant degradation in quality.

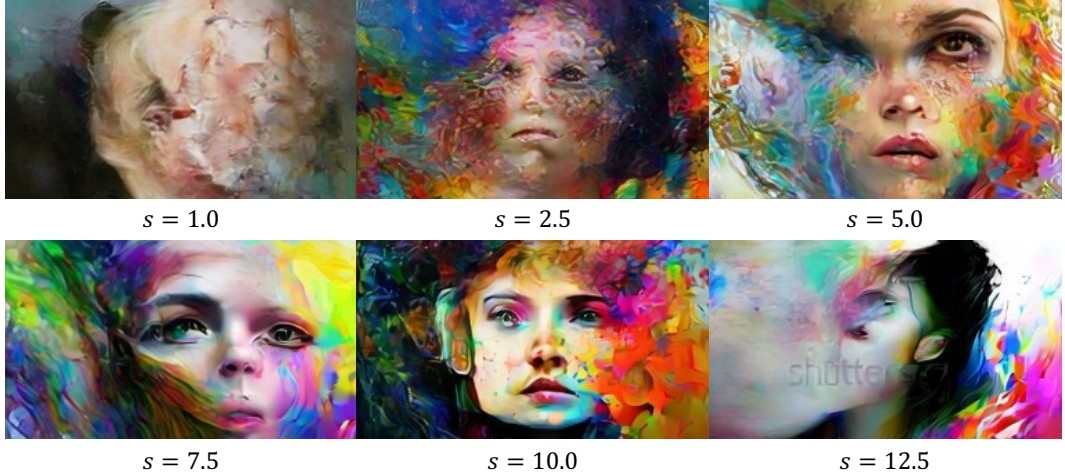

Figure 8: The influence of classifier free guidance $s$.

**Classifier-free guidance.** An appropriate choice of the CFG scale can significantly improve generation quality. As illustrated in Fig. 8, setting $s$ in the range of 5–7.5 leads to more complete and coherent images. However, further increasing the CFG value reduces realism and may even introduce artifacts such as watermarks inherited from the training data.

**Denoising steps.** The number of iterative steps in the denoising head has a substantial impact on generation quality. As shown in Fig. 9, meaningful images can be produced with as few as 10 steps,

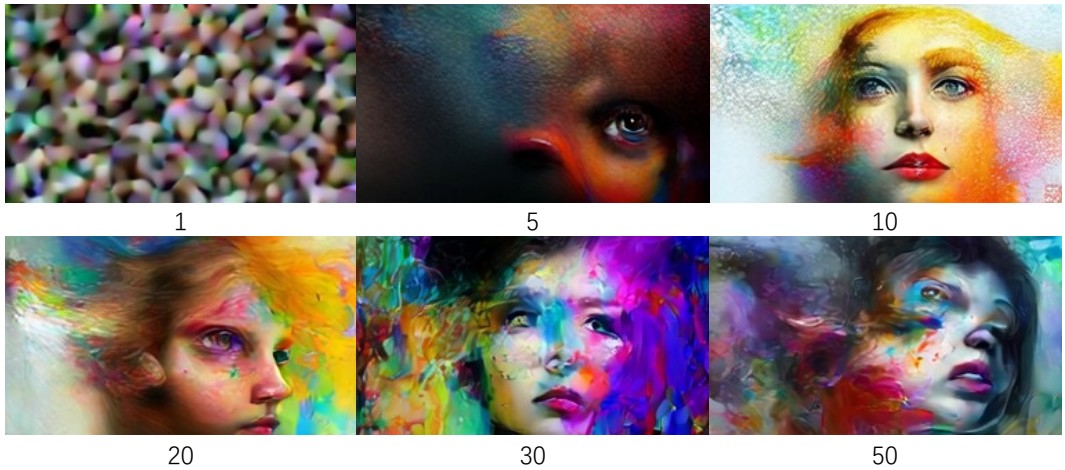

|  | 1 | 5 | 10 |
|  | 20 | 30 | 50 |

Figure 9: The influence of denosing steps in denosing heads.

|  | OMNIAR-I | OMNIAR-IV | OMNIAR-AVI |
|---|---|---|---|
| *Parameters dsitribution* | | | |
| Vocabulary mapping | 155M | 311M | 388M |
| Omni-encoder | 353M | 355M | 355M |
| MoT | 362M+362M | 1.25B+1.25B | 3.4B+3.4B |
| Image denoising head | 309M | 312M | 313M |
| Video denoising head | N/A | 329M | 329M |
| Audio denoising head | N/A | N/A | 313M |
| LLM head | 155M | 311M | 388M |
| Total params | 1.3B+362M | 2.9B+1.25B | 5.48B+3.4B |
| *Training time distribution* | | | |
| VAE encoding | 1.3% | 3.4% | 10.8% |
| Omni-encoder forward | 3.2% | 6.3% | 12.4% |
| Omni-encoder EMA forward | 2.4% | 2.4% | 1.2% |
| MoT forward | 22.5% | 25.3% | 22.3% |
| Backward | 53.9% | 47.4% | 42.3% |
| Update params | 16.5% | 14.9% | 11.0% |
| *Training modal distribution* | | | |
| Text | 22.2% | 9.3% | 5.7% |
| Image | 77.8% | 45.4% | 34.9% |
| Video | N/A | 45.2% | 44.9% |
| Audio | N/A | N/A | 14.5% |

Table 3: Model parameters, computation, and modality distribution across different model scales.

which highlights the potential for real-time generation. We find that 20–30 steps typically yield the most visually appealing results. However, blindly increasing the number of denoising steps does not consistently improve image quality; for example, using 50 steps often introduces local distortions. Note that we omit results on varying AR steps during generation, since the number of groups is fixed according to Eq. 1, consistent with the training setup. This design reduces hyperparameter tuning during sampling and stabilizes the overall performance.

## C   IMPLEMENTATION DETAILS

Our implementation is guided by three central design choices. First, we emphasize *temporal alignment* across modalities to ensure consistent synchronization between audio, video, and text. Second, we adopt a *progressive scaling strategy* in which model capacity and modality coverage grow coherently

with scale. Third, we rely on a *unified joint training protocol* that integrates multimodal data at scale while balancing efficiency and performance. We describe these design choices in detail below.

## C.1 MODEL DESIGN

**Tokenization and Patchification.** Text inputs are processed using the QWEN3 tokenizer, which produces discrete token IDs mapped to dense embeddings through a single embedding layer. Continuous modalities such as images, videos, and audio are encoded by compact KL-VAE encoders that yield dense latent feature maps suitable for patchification. After modality-specific patchification, all latent representations are projected into a unified sequence of *omni-tokens* through the shared omni-encoder.

For video, an input clip of $4T+1$ frames is reduced to $T+1$ latent frames by WANX's KL-VAE (Wan et al., 2025) with a temporal stride of $s = 4$. This yields 4 latent frames per second at 16 FPS, with spatial downsampling by a factor of 8 and expansion of the channel dimensionality from 3 (RGB) to 16. Images undergo the same spatial encoding process. Audio waveforms are converted into mel spectrograms (16 kHz sampling rate, hop size of 256, 80 mel bins), then downsampled by a factor of 2 and compressed into latent maps with 20 channels, resulting in 32 time steps per second, as depicted in Huang et al. (2023).

A key design principle is temporal alignment. Audio produces 32 latent steps per second, while video produces 4 latent frames per second. To reconcile them, we use an audio patch size of 8 and a video patch size of 1, resulting in 4 tokens per second for both modalities. Each token therefore corresponds to 0.25 seconds, enabling precise synchronization. After omni-encoding, audio and video tokens are interleaved to strengthen alignment without regrouping, following designs in QWEN-OMNI (Xu et al., 2025). For spatial patchification of images and frames, we adopt a standard patch size of 2 (Peebles & Xie, 2023).

**Model Configurations.** The smallest model, OMNIAR-I, contains 0.6B parameters and supports image-only understanding and generation. A medium-scale model, OMNIAR-IV, with 1.7B parameters, extends to video, enabling image+video understanding, generation, and image-to-video synthesis. The largest model, OMNIAR-AVI, with 4B parameters, incorporates audio, supporting fully integrated multimodal tasks such as synchronized audio–video generation and audio-driven video synthesis, including speech-driven talking heads and ambient sound generation.

## C.2 MULTI-TIMBRE GENERATION

Unlike image and video synthesis, audio generation often requires modeling multiple timbres. Our general strategy is to assign distinct *modal embeddings* to indicate different timbres, while sharing a single audio denoising head. In TTS, for example, each speaker is assigned a unique modal embedding to represent their specific timbre. For audio–video generation, background sounds without human voices are associated with a separate modal embedding. When synthesizing speech mixed with natural background sounds, we simultaneously use the speaker's embedding and the background embedding to generate the two streams of audio before mixing them. Additionally, for unclassified sounds, we employ a default modal embedding. This embedding allocation strategy enables the model to flexibly and effectively support multi-timbre audio generation.

Although OMNIAR-AVI in this work adopts a continuous KL-VAE for audio encoding followed by a shared omni-encoder, we observe a substantial modality gap between speech and vision. Directly sharing the omni-encoder with visual features degrades the quality of both image and video generation. Meanwhile, recent studies have shown strong performance in audio generation using discrete audio tokens. We hypothesize that this is because audio, like text, is inherently one-dimensional and thus well-suited to discrete representations. Consequently, in future work we plan to adopt a discrete audio tokenizer, similar to Boson AI (2025), to encode audio signals. This approach is expected to alleviate the degradation of visual quality caused by sharing the omni-encoder.

## C.3 TRAINING DETAILS

**Datasets and Preprocessing.** Image data are sourced from LAION-5B (Schuhmann et al., 2022), COCO (Lin et al., 2014), COYO-700M (Byeon et al., 2022), and JourneyDB (Sun et al., 2023a),

with captions regenerated using InternVL-2.5 (Chen et al., 2024e) and LLaVA (Liu et al., 2024a). Video data are drawn from Panda70M (Chen et al., 2024d), HD-VG130M (Wang et al., 2023), and WebVid10M (Bain et al., 2021), segmented into 4–10 second intervals with descriptions generated by Tarsier (Wang et al., 2024a). Audio is extracted from video soundtracks, focusing on speech segments or clips with visible speakers for audio-driven generation. Images are resized to a short edge within $[144, 256]$ and capped at $256 \times 256$. Videos are resized to $144 \times 256$ (or $256 \times 144$) and clipped to 4–8 seconds. Audio is trimmed or padded to match paired video lengths before mel conversion. While simple operations such as random horizontal flipping accelerate convergence, we also adopt a randomized augmentation strategy inspired by Yu et al. (2024). At each training step, a random starting token is chosen, and subsequent neighboring tokens are generated deterministically within its local context. This randomized-local augmentation enhances multimodal comprehension while preserving generative quality.

**Training Protocol.**   Overall, we find that the difficulty of learning generative tasks varies across modalities: audio is the easiest, followed by motion information, while the most challenging is generating images from scratch. Therefore, in multimodal training for OMNIAR-IV and OMNIAR-IVA, we always ensure that text-to-image tasks account for at least 50% of training. To support CFG sampling, prompts are randomly dropped with probability 0.3. Captioning is considered relatively simple, so only 10% of training samples are assigned to caption tasks. Notably, during video generation, the first frame is always trained as an image, split into multiple groups and rendered using the image denoising head. Subsequent frames are conditioned on preceding frames, focusing on motion generation. Optimization is performed using AdamW (Loshchilov et al., 2017) ($\beta_1 = 0.9, \beta_2 = 0.95, \epsilon = 1 \times 10^{-15}$) with zero weight decay. Based on the model size and memory allocation strategy, we predefine the maximum training sequence length that can be supported. During sampling, data from the same task are cached until they can be assembled into a batch with consistent dimensions (*e.g.*, identical image sizes or equal video lengths). Once the maximum sequence length is reached, the samples are combined into a batch for training. Unlike the sequence packing strategy commonly adopted in LLM training, we deliberately employ this batching scheme, which proves to be more efficient in our multimodal setting. The learning rate warms up linearly from $1 \times 10^{-7}$ to $1 \times 10^{-4}$ over 5k steps and then remains constant. Training proceeds for 1M steps with an EMA factor of 0.9999. Gradients are clipped at 1.0. Efficiency is further improved with gradient checkpointing and FSDP hybrid sharding, with optional CPU offloading (Paszke et al., 2019).

As shown in Tab. 3, we report the model parameters and training performance. Notably, the additional overhead of Omni-encoder EMA forward arises from the use of the JEPA loss. However, this overhead remains below 3% across all model scales, while JEPA training consistently accelerates convergence, making it a worthwhile trade-off.

# D   TRAINING ANALYSIS

Since few prior works adopt designs similar to OMNIAR—including self-token prediction and JEPA training within LLMs—we provide convergence curves across different scales and modalities in Fig. 10. We observe that the flow matching loss quickly converges to a small value and then decreases slowly and stably, consistent with findings in diffusion and flow matching models. Notably, halving this loss does not necessarily indicate satisfactory generation quality: around 10k steps the model can move beyond pure noise and produce image patches, around 40k it can generate simple single-object images, while more complex or rare cases require training beyond 300k steps (for OMNIAR-I; larger models converge faster). Text prediction loss also drops rapidly and remains low. In contrast, JEPA loss exhibits a different trajectory: it rises sharply from near zero to a peak, then gradually decreases before slightly increasing and stabilizing at a moderate value—behavior aligned with its adversarial nature of balancing generation and understanding. Excessive weighting of $\lambda_{\text{JEPA}}$ prevents effective training, as shown in Fig. 10h, where the loss keeps shrinking without improving generation quality despite decreasing flow matching loss. Finally, Fig. 11 shows that gradient magnitudes shrink with model scale, indicating stronger learning capacity, while increasing modalities slightly reduces stability (*e.g.*, an outlier appears near 3k steps for OMNIAR-AVI).

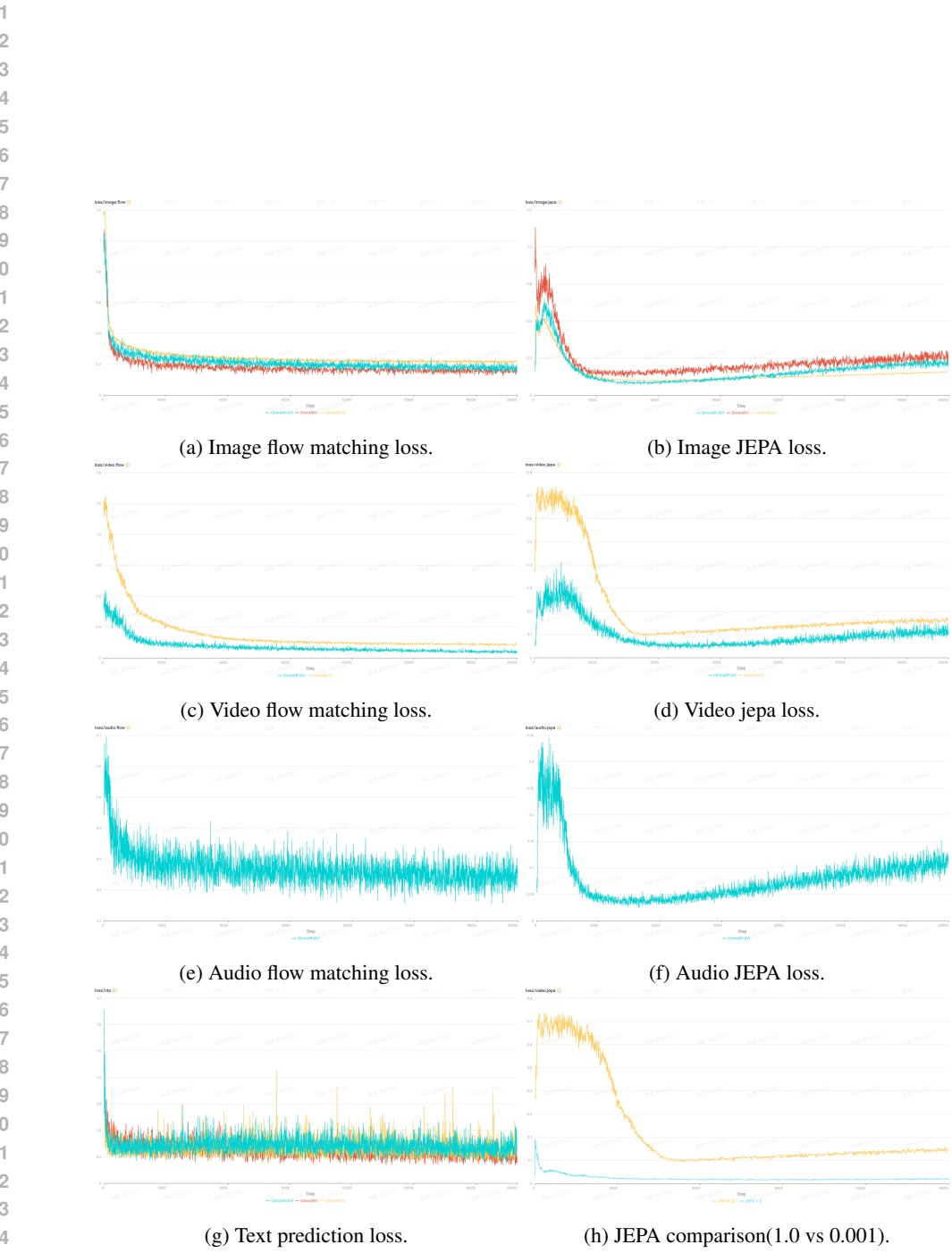

(a) Image flow matching loss.

(b) Image JEPA loss.

(c) Video flow matching loss.

(d) Video jepa loss.

(e) Audio flow matching loss.

(f) Audio JEPA loss.

(g) Text prediction loss.

(h) JEPA comparison(1.0 vs 0.001).

Figure 10: Training curve.

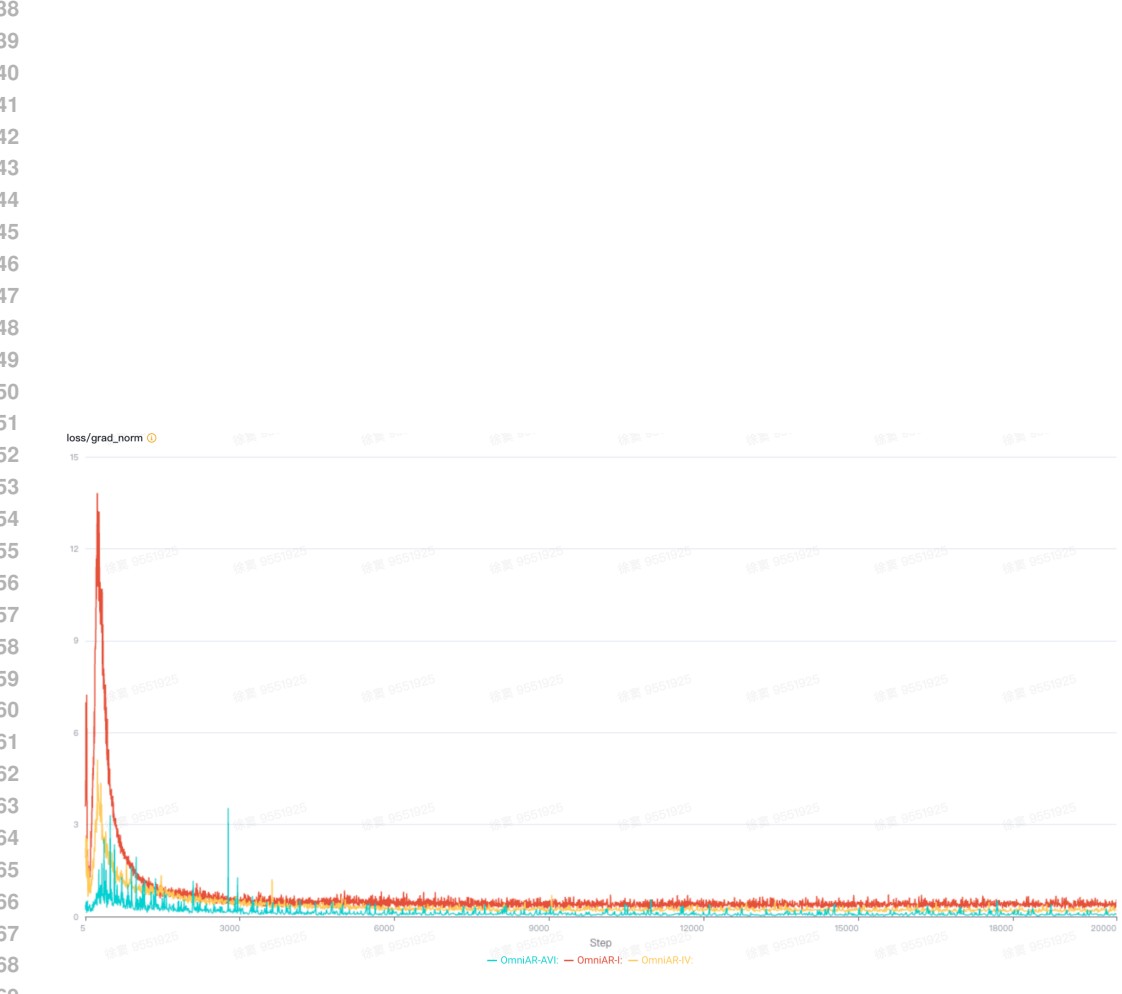

Figure 11: Gradient norm curve.

