# OpenReview forum: "Autoregression with Self-Token Prediction"
_ICLR.cc/2026/Conference — ICLR 2026 Conference Withdrawn Submission_

### Official Review · Reviewer_KPNS · 2025-10-30

**Soundness:** 3
**Presentation:** 3
**Contribution:** 3
**Rating:** 4
**Confidence:** 5

**Summary:**

The paper proposes self-token prediction for omnimodal autoregressive modeling, addressing the tendency of next-token regression on continuous latents to collapse into identity copying via a two-stream, asymmetric attention design where only reference tokens serve as context. The approach is paired with grouped causal attention to maintain causality while enabling parallel token prediction within groups, a JEPA regularizer that aligns causal features to a full-context EMA teacher across modalities, and a lightweight time reparameterization schedule to stabilize token-level flow sampling. The resulting system (OMNIAR) aims to retain KV-cache–friendly, causal decoding while scaling to text, images, audio, and video. Empirical evaluation covers ImageNet conditional image generation and classification (reporting FID/IS/Accuracy), text-to-image (GenEval), image captioning on COCO via BLIP, and video generation with VBench.

**Strengths:**

- Proposed to adopt multiple techniques to improve performance, such as grouped causal attention that allows parallel token prediction, JEPA loss and a lightweight time reparameterization schedule.
- The model is causal, autoregressive and KV-cache–friendly.

**Weaknesses:**

- Line 76-78: “although the fraction of masked tokens is sampled stochastically during training(see Fig. 1, middle), each forward pass uses only a single sampled mask ratio; consequently, training sees only isolated masking configurations rather than the progressive unmasking dynamics that occur during generation.” The authors claim that masked-based models have train-inference mismatch due to a single sampled mask ratio for the sequence. However, I want to point out that each sequence will be applied masking with different masking ratios in different training iterations, so the model is able to see the same sequence with different masks throughout the training process.
- The attention map in Fig. 3 is unclear which tokens belong to the same group for grouped causal attention.
- Table 1 (a) is a system-level comparison with other models, however, the authors only compared with models that have small model size. In table 2, it seems that models are able to scale up to 5.5B. Why not also scale up the proposed models in table 1 for class-conditional image generation and compare between the model with self-token prediction and MAR-H (FID: 1.55 w/ CFG)?
- No inference speed analyses in this work, although adopting techniques like parallel token prediction and kv-caching. A small table with throughput/latency vs. other system-level baselines would strengthen the empirical case.
- In Eq. (9), the ∆tt notation is confusing. It should be written as $t \cdot \delta t$.

Lacking references to:
MELLE [A], IMPACT [B], AudioMNTP [C], and GLM-4-voice [D], which are all autoregressive speech/audio generation models.

[A] Meng, Lingwei, et al. "Autoregressive Speech Synthesis without Vector Quantization." CoRR (2024).

[B] Huang, KP, et al. "IMPACT: Iterative Mask-based Parallel Decoding for Text-to-Audio Generation with Diffusion Modeling." ICML 2025

[C] Yang, SW, et al. "Generative Audio Language Modeling with Continuous-valued Tokens and Masked Next-Token Prediction." ICML 2025

[D] Zeng, Aohan, et al. "Glm-4-voice: Towards intelligent and human-like end-to-end spoken chatbot." arXiv preprint arXiv:2412.02612 (2024).

**Questions:**

Why does JEPA loss encourage modality alignment? Is it because tokens of each modality use the same loss and hence are projected to the same target space?

With the linear interpolation you use, do the authors agree that Eq. (5) is equivalent to rectified flow. If the authors agree with this, please cite rectified flow’s original paper [E].

[E] Liu, Xingchao, Chengyue Gong, and Qiang Liu. "Flow straight and fast: Learning to generate and transfer data with rectified flow." arXiv preprint arXiv:2209.03003 (2022).

---

### Official Review · Reviewer_uyCc · 2025-10-31

**Soundness:** 4
**Presentation:** 2
**Contribution:** 3
**Rating:** 4
**Confidence:** 3

**Summary:**

The paper introduces "Self-Token Prediction," a novel paradigm designed to overcome the limitations of next-token prediction and masked prediction for continuous data (images, video, audio). The core idea is to condition the prediction of each token on ground-truth "reference" tokens during training, using an asymmetric grouped causal attention mechanism. This design prevents the model from learning an identity function and ensures training-inference consistency.

**Strengths:**

The "Self-Token Prediction" paradigm is a genuine innovation. It elegantly solves the identity collapse problem in continuous autoregression and the train-test mismatch of masking, two significant challenges in the field.

**Weaknesses:**

1. The explanation of the Self-Token Prediction, which is the paper's core contribution, is not sufficiently clear. Figure 3 (right) is critical for understanding the information flow between reference and prediction tokens but lacks detailed annotations.
2. The experimental proof is currently insufficient to fully support the paper's ambitious motivation of building a unified model for "omnimodal AR understanding and generation." This is the most significant shortcoming of this paper. There is a notable lack of quantitative results for: text-video, text-audio, and image-audio generation. Quantitative results for video captioning and audio captioning are absent.
The current experiments leans heavily on image, leaving the "omni" claim only partially validated.
3. What is "KL-VAE"? I dont know which previous paper introduces this item.

**Questions:**

See Weaknesses

---

### Official Review · Reviewer_g7Xx · 2025-11-01

**Soundness:** 2
**Presentation:** 1
**Contribution:** 1
**Rating:** 2
**Confidence:** 4

**Summary:**

The paper presents OmniAR, a unified generative framework extending next-token prediction to continuous modalities such as images, audio, and video. It introduces self-token prediction, where each token is conditioned on a ground-truth reference during training, preserving causal consistency and avoiding identity collapse. This approach supports key–value caching and parallel generation, enabling efficient, scalable, and high-fidelity synthesis across multiple modalities, including real-time and theoretically endless video generation.

**Strengths:**

1. Introduction of a unified self-token prediction framework.
- OmniAR proposes a new training paradigm in which each token receives an explicit reference token during training, effectively bridging autoregressive and mask-based prediction.

**Weaknesses:**

1. Insufficient ablation and comparative analysis.
- The paper introduces a new generative modeling paradigm, but the ablation studies are limited. In Section 5.1 (Efficiency of Self-Token Prediction), it would be valuable to include a direct comparison among Next-token prediction, Mask-based prediction, and Self-token prediction on a common benchmark such as ImageNet. Furthermore, Equation (6) introduces an additional JEPA loss, yet there is no ablation isolating its contribution, making it difficult to assess how much this component improves the final performance.


2. Lack of theoretical and empirical justification for “real-time and endless video generation.”
- The claim regarding real-time and theoretically endless video generation is insufficiently supported. Stronger evidence is needed to demonstrate concrete inference efficiency gains, such as reductions in sampling steps or wall-clock time compared with existing baselines. Moreover, if the proposed approach improves long-sequence generation, experiments and quantitative evaluations on long video generation should be provided to substantiate the claim.


3. Limited discussion of related generative frameworks.
- The experiment section could be expanded to discuss several recent generative modeling frameworks that are closely connected to this work, including ACDiT [1], GIVT [2], MAGViT-2 [3] and ResGen [4]. Including these work in Table 1 would help position OmniAR more clearly within the broader landscape of contemporary generative modeling research.


References

[1] Li et al., ACDiT: Interpolating Autoregressive Conditional Modeling and Diffusion Transformer, 2025.

[2] Tschannen et al., Generative Infinite-Vocabulary Transformers, 2024.

[3] Yu et al., Language Model Beats Diffusion -- Tokenizer is Key to Visual Generation, 2024

[4] Kim et al., Efficient Generative Modeling with Residual Vector Quantization-Based Tokens, 2025.

**Questions:**

N/A

---

### Note · Authors · 2025-11-12

I have read and agree with the venue's withdrawal policy on behalf of myself and my co-authors.